# A Panda? No, It's a Sloth: Slowdown Attacks on Adaptive Multi-Exit Neural Network Inference

**Sanghyun Hong**[*]**, Yiğitcan Kaya**[*]**, Ionuț-Vlad Modoranu**[†]**, Tudor Dumitraş**
University of Maryland, College Park, USA
[†]Alexandru Ioan Cuza University, Iași, Romania
shhong@cs.umd.edu, yigitcan@cs.umd.edu,
modoranu.ionut.vlad@hotmail.com, tudor@umd.edu

## ABSTRACT

Recent increases in the computational demands of deep neural networks (DNNs), combined with the observation that most input samples require only simple models, have sparked interest in *input-adaptive* multi-exit architectures, such as MSDNets or Shallow-Deep Networks. These architectures enable faster inferences and could bring DNNs to low-power devices, *e.g.*, in the Internet of Things (IoT). However, it is unknown if the computational savings provided by this approach are robust against adversarial pressure. In particular, an adversary may aim to slowdown adaptive DNNs by increasing their average inference time—a threat analogous to the *denial-of-service* attacks from the Internet. In this paper, we conduct a systematic evaluation of this threat by experimenting with three generic multi-exit DNNs (based on VGG16, MobileNet, and ResNet56) and a custom multi-exit architecture, on two popular image classification benchmarks (CIFAR-10 and Tiny ImageNet). To this end, we show that adversarial example-crafting techniques can be modified to cause slowdown, and we propose a metric for comparing their impact on different architectures. We show that a slowdown attack reduces the efficacy of multi-exit DNNs by 90–100%, and it amplifies the latency by 1.5–5× in a typical IoT deployment. We also show that it is possible to craft universal, reusable perturbations and that the attack can be effective in realistic black-box scenarios, where the attacker has limited knowledge about the victim. Finally, we show that adversarial training provides limited protection against slowdowns. These results suggest that further research is needed for defending multi-exit architectures against this emerging threat. Our code is available at https://github.com/sanghyun-hong/deepsloth.

## 1 INTRODUCTION

The inference-time computational demands of deep neural networks (DNNs) are increasing, owing to the "going deeper" (Szegedy et al., 2015) strategy for improving accuracy: as a DNN gets deeper, it progressively gains the ability to learn higher-level, complex representations. This strategy has enabled breakthroughs in many tasks, such as image classification (Krizhevsky et al., 2012) or speech recognition (Hinton et al., 2012), at the price of costly inferences. For instance, with 4× more inference cost, a 56-layer ResNet (He et al., 2016) improved the Top-1 accuracy on ImageNet by 19% over the 8-layer AlexNet. This trend continued with the 57-layer state-of-the-art EfficientNet (Tan & Le, 2019): it improved the accuracy by 10% over ResNet, with 9× costlier inferences.

The accuracy improvements stem from the fact that the deeper networks *fix* the mistakes of the shallow ones (Huang et al., 2018). This implies that some samples, which are already correctly classified by shallow networks, do not necessitate the extra complexity. This observation has motivated research on *input-adaptive* mechanisms, in particular, multi-exit architectures (Teerapittayanon et al., 2016; Huang et al., 2018; Kaya et al., 2019; Hu et al., 2020). Multi-exit architectures save computation by making input-specific decisions about bypassing the remaining layers, once the model becomes confident, and are orthogonal to techniques that achieve savings by permanently modifying the

---

[*]Authors contributed equally.

model (Li et al., 2016; Banner et al., 2018; Han et al., 2015; Taylor et al., 2018). Figure 1 illustrates how a multi-exit model (Kaya et al., 2019), based on a standard VGG-16 architecture, correctly classifies a selection of test images from 'Tiny ImageNet' before the final layer. We see that more typical samples, which have more supporting examples in the training set, require less depth and, therefore, less computation.

It is unknown if the computational savings provided by multi-exit architectures are robust against adversarial pressure. Prior research showed that DNNs are vulnerable to a wide range of attacks, which involve imperceptible input perturbations (Szegedy et al., 2014; Goodfellow et al., 2015; Papernot et al., 2016; Hu et al., 2020). Considering that a multi-exit model, on the worst-case input, does not provide any computational savings, we ask: *Can the savings from multi-exit models be maliciously negated by input perturbations?* As some natural inputs do require the full depth of the model, it may be possible to craft adversarial examples that delay the correct decision; it is unclear, however, how many inputs can be delayed with imperceptible perturbations. Furthermore, it is unknown if universal versions of these adversarial examples exist, if the examples transfer across multi-exit architectures and datasets, or if existing defenses (*e.g.* adversarial training) are effective against slowdown attacks.

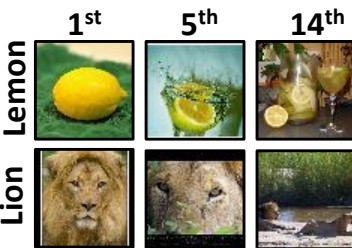

Figure 1: **Simple to complex inputs.** Some Tiny ImageNet images a VGG-16 model can correctly classify if computation stops at the $1^{st}$, $5^{th}$, and $14^{th}$ layers.

**Threat Model.** We consider a new threat against DNNs, analogous to the *denial-of-service (DoS) attacks* that have been plaguing the Internet for decades. By imperceptibly perturbing the input to trigger this worst-case, the adversary aims to slow down the inferences and increase the cost of using the DNN. This is an important threat for many practical applications, which impose strict limits on the responsiveness and resource usage of DNN models (*e.g.* in the Internet-of-Things (Taylor et al., 2018)), because the adversary could push the victim outside these limits. For example, against a commercial image classification system, such as Clarifai.com, a slowdown attack might waste valuable computational resources. Against a model partitioning scheme, such as Big-Little (De Coninck et al., 2015), it might introduce network latency by forcing excessive transmissions between local and remote models. A *slowdown attack* aims to force the victim to do more work than the adversary, e.g. by amplifying the latency needed to process the sample or by crafting reusable perturbations. The adversary may have to achieve this with incomplete information about the multi-exit architecture targeted, the training data used by the victim or the classification task (see discussion in Appendix A).

**Our Contributions.** To our best knowledge, we conduct the first study of the robustness of multi-exit architectures against adversarial slowdowns. To this end, we find that examples crafted by prior evasion attacks (Madry et al., 2017; Hu et al., 2020) fail to bypass the victim model's early exits, and we show that an adversary can adapt such attacks to the goal of model slowdown by modifying its objective function. We call the resulting attack *DeepSloth*. We also propose an *efficacy* metric for comparing slowdowns across different multi-exit architectures. We experiment with three generic multi-exit DNNs (based on VGG16, ResNet56 and MobileNet) (Kaya et al., 2019) and a specially-designed multi-exit architecture, MSDNets (Huang et al., 2018), on two popular image classification benchmarks (CIFAR-10 and Tiny ImageNet). We find that DeepSloth reduces the efficacy of multi-exit DNNs by 90–100%, *i.e.*, the perturbations render nearly *all early exits* ineffective. In a scenario typical for IoT deployments, where the model is partitioned between edge devices and the cloud, our attack amplifies the latency by 1.5–5×, negating the benefits of model partitioning. We also show that it is possible to craft a universal DeepSloth perturbation, which can slow down the model on either all or a class of inputs. While more constrained, this attack still reduces the efficacy by 5–45%. Further, we observe that DeepSloth can be effective in some black-box scenarios, where the attacker has limited knowledge about the victim. Finally, we show that a standard defense against adversarial samples—adversarial training—is inadequate against slowdowns. Our results suggest that further research will be required for protecting multi-exit architectures against this emerging security threat.

## 2 RELATED WORK

**Adversarial Examples and Defenses.** Prior work on adversarial examples has shown that DNNs are vulnerable to test-time input perturbations (Szegedy et al., 2014; Goodfellow et al., 2015; Papernot et al., 2017; Carlini & Wagner, 2017; Madry et al., 2018). An adversary who wants to maximize a model's error on specific test-time samples can introduce human-imperceptible perturbations to these samples. Moreover, an adversary can also exploit a *surrogate* model for launching the attack and still hurt an unknown victim (Athalye et al., 2018; Tramèr et al., 2017b; Inkawhich et al., 2019). This *transferability* leads to adversarial examples in more practical black-box scenarios. Although many defenses (Kurakin et al., 2016; Xu et al., 2017; Song et al., 2018; Liao et al., 2018; Lecuyer et al., 2019) have been proposed against this threat, *adversarial training* (AT) has become the frontrunner (Madry et al., 2018). In Sec 5, we evaluate the vulnerability of multi-exit DNNs to adversarial slowdowns in white-box and black-box scenarios. In Sec 6, we show that standard AT and its simple adaptation to our perturbations are not sufficient for preventing slowdown attacks.

**Efficient Input-Adaptive Inference.** Recent input-adaptive DNN architectures have brought two seemingly distant goals closer: achieving both high predictive quality and computational efficiency. There are two types of input-adaptive DNNs: adaptive neural networks (AdNNs) and multi-exit architectures. During the inference, AdNNs (Wang et al., 2018; Figurnov et al., 2017) dynamically skip a certain part of the model to reduce the number of computations. This mechanism can be used only for ResNet-based architectures as they facilitate skipping within a network. On the other hand, multi-exit architectures (Teerapittayanon et al., 2016; Huang et al., 2018; Kaya et al., 2019) introduce multiple side branches—or early-exits—to a model. During the inference on an input sample, these models can preemptively stop the computation altogether once the stopping criteria are met at one of the branches. Kaya et al. (2019) have also identified that standard, non-adaptive DNNs are susceptible to *overthinking*, *i.e.*, their inability to stop computation leads to inefficient inferences on many inputs.

Haque et al. (2020) presented attacks specifically designed for reducing the energy-efficiency of AdNNs by using adversarial input perturbations. However, our work studies a new threat model that an adversary causes slowdowns on multi-exit architectures. By imperceptibly perturbing the inputs, our attacker can (i) introduce network latency to an infrastructure that utilizes multi-exit architectures and (ii) waste the victim's computational resources. To quantify this vulnerability, we define a new metric to measure the impact of adversarial input perturbation on different multi-exit architectures (Sec 3). In Sec 5, we also study practical attack scenarios and the transferability of adversarial input perturbations crafted by our attacker. Moreover, we discuss the potential defense mechanisms against this vulnerability, by proposing a simple adaptation of adversarial training (Sec 6). To the best of our knowledge, our work is the first systematic study of this new vulnerability.

**Model Partitioning.** Model partitioning has been proposed to bring DNNs to resource-constrained devices (De Coninck et al., 2015; Taylor et al., 2018). These schemes split a multi-exit model into sequential components and deploy them in separate endpoints, e.g., a small, local on-device part and a large, cloud-based part. For bringing DNNs to the Internet of Things (IoT), partitioning is instrumental as it reduces the transmissions between endpoints, a major bottleneck. In Sec 5.1, on a partitioning scenario, we show that our attack can force excessive transmissions.

## 3 EXPERIMENTAL SETUP

**Datasets.** We use two datasets: CIFAR-10 (Krizhevsky et al., 2009) and Tiny-ImageNet (Tiny). For testing the cross-domain transferability of our attacks, we use the CIFAR-100 dataset.

**Architectures and Hyper-parameters.** To demonstrate that the vulnerability to adversarial slowdowns is common among multi-exit architectures, we experiment on two recent techniques: Shallow-Deep Networks (SDNs) (Kaya et al., 2019) and MSDNets (Huang et al., 2018). These architectures were designed for different purposes: SDNs are generic and can convert any DNN into a multi-exit model, and MSDNets are custom designed for efficiency. We evaluate an MSDNet architecture (6 exits) and three SDN architectures, based on VGG-16 (Simonyan & Zisserman, 2014) (14 exits), ResNet-56 (He et al., 2016) (27 exits), and MobileNet (Howard et al., 2017) (14 exits).

**Metrics.** We define the *early-exit capability* (EEC) curve of a multi-exit model to indicate the fraction of the test samples that exit early at a specific fraction of the model's full inference cost.

Figure 2 shows the EEC curves of our SDNs on Tiny ImageNet, assuming that the computation stops when there is a correct classification at an exit point. For example, VGG-16-based SDN model can correctly classify ∼50% of the samples using ∼50% of its full cost. Note that this stopping criterion is impractical; in Sec 4, we will discuss the practical ones.

We define the early-exit efficacy, or *efficacy* in short, to quantify a model's ability of utilizing its exit points. The efficacy of a multi-exit model is the area under its EEC curve, estimated via the trapezoidal rule. An ideal efficacy for a model is close to $1$, when most of the input samples the computation stops very early; models that do not use their early exits have $0$ efficacy. A model with low efficacy generally exhibits a higher *latency*; in a partitioned model, the low efficacy will cause more input transmissions to the cloud, and the latency is further *amplified* by the network round trips. A multi-exit model's efficacy and accuracy are dictated by its stopping criteria, which we discuss in the next section. As for the classification performance, we report the Top-1 accuracy on the test data.

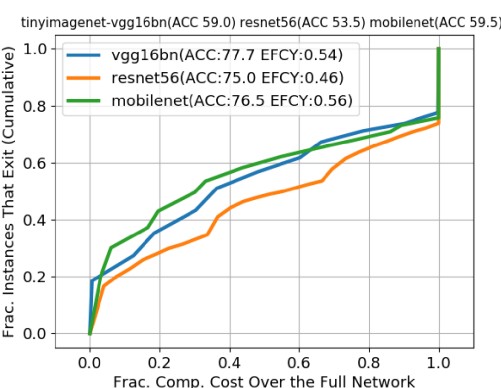

Figure 2: **The EEC curves.** Each curve shows the fraction of test samples a model classifies using a certain fraction of its full inference cost. 'EFCY' is short for the model's efficacy.

## 4    ATTACKING THE MULTI-EXIT ARCHITECTURES

**Setting.** We consider the supervised classification setting with standard feedforward DNN architectures. A DNN model consists of $N$ blocks, or layers, that process the input sample, $x \in \mathbb{R}^d$, from beginning to end and produce a classification. A classification, $F(x, \theta) \in \mathbb{R}^m$, is the predicted probability distribution of $x$ belonging to each label $y \in M = \{1, ..., m\}$. Here, $\theta$ denotes the tunable parameters, or the weights, of the model. The parameters are learned on a training set $\mathcal{D}$ that contains multiple $(x_i, y_i)$ pairs; where $y_i$ is the ground-truth label of the training sample $x_i$. We use $\theta_i$ to denote the parameters at and before the $i^{th}$ block; *i.e.*, $\theta_i \subset \theta_{i+1}$ and $\theta_N = \theta$. Once a model is trained, its performance is then tested on a set of unseen samples, $\mathcal{S}$.

**Multi-Exit Architectures.** A multi-exit model contains $K$ *exit points*—internal classifiers—attached to a model's hidden blocks. We use $F_i$ to denote the $i^{th}$ exit point, which is attached to the $j^{th}$ block. Using the output of the $j^{th}$ ($j < N$) block on $x$, $F_i$ produces an internal classification, i.e., $F_i(x, \theta_j)$, which we simply denote as $F_i(x)$. In our experiments, we set $K = N$ for SDNs, *i.e.*, one internal classifier at each block and $K = 6$ for MSDNets. Given $F_i(x)$, a multi-exit model uses deterministic criteria to decide between forwarding $x$ to compute $F_{i+1}(x)$ and stopping for taking the *early-exit* at this block. Bypassing early-exits decreases a network's efficacy as each additional block increases the inference cost. Note that multi-exit models process each sample individually, not in batches.

**Practical Stopping Criteria.** Ideally, a multi-exit model stops when it reaches a correct classification at an exit point, *i.e.*, $\text{argmax}_{j \in M} F_i^{(j)}(x) = \hat{y}_i = y$; $y$ is the ground-truth label. However, for unseen samples, this is impractical as $y$ is unknown. The prior work has proposed two simple strategies to judge whether $\hat{y}_i = y$: $F_i(x)$'s entropy (Teerapittayanon et al., 2016; Huang et al., 2018) or its confidence (Kaya et al., 2019). Our attack (see Sec 4.3) leverages the fact that a uniform $F_i(x)$ has both the highest entropy and the lowest confidence. For generality, we experiment with both confidence-based—SDNs—and entropy-based—MSDNets—strategies.

A strategy selects confidence, or entropy, thresholds, $T_i$, that determine whether the model should take the $i^{th}$ exit for an input sample. Conservative $T_i$'s lead to fewer early exits and the opposite hurts the accuracy as the estimate of whether $\hat{y}_i = y$ becomes unreliable. As utility is a major practical concern, we set $T_i$'s for balancing between efficiency and accuracy. On a holdout set, we set the thresholds to maximize a model's efficacy while keeping its relative accuracy drop (RAD) over its maximum accuracy within 5% and 15%. We refer to these two settings as RAD<5% and RAD<15%. Table 2 (*first* segment) shows how accuracy and efficacy change in each setting.

## 4.1 THREAT MODEL

We consider an adversary who aims to decrease the early-exit efficacy of a *victim* model. The attacker crafts an *imperceptible* adversarial perturbation, $v \in \mathbb{R}^d$ that, when added to a test-time sample $x \in \mathcal{S}$, prevents the model from taking early-exits.

**Adversary's Capabilities.** The attacker is able to modify the victim's test-time samples to apply the perturbations, *e.g.*, by compromising a camera that collects the data for inference. To ensure the imperceptibility, we focus on $\ell_\infty$ norm bounded perturbations as they (i) are well are studied; (ii) have successful defenses (Madry et al., 2018); (iii) have prior extension to multi-exit models (Hu et al., 2020); and (iv) are usually the most efficient to craft. We show results on $\ell_2$ and $\ell_1$ perturbations in Appendix C. In line with the prior work, we bound the perturbations as follows: for CIFAR-10, $||v||_\infty \leq \epsilon = 0.03$ (Madry et al., 2017), $||v||_1 \leq 8$ (Tramèr & Boneh, 2019) and $||v||_2 \leq 0.35$ (Chen et al., 2017); for Tiny ImageNet, $||v||_\infty \leq \epsilon = 0.03$ (Yang et al., 2019), $||v||_1 \leq 16$ and $||v||_2 \leq 0.6$.

**Adversary's Knowledge.** To assess the security vulnerability of multi-exit architectures, we study *white-box scenarios*, *i.e.*, the attacker knows all the details of the victim model, including its $\mathcal{D}$ and $\theta$. Further, in Sec 5.2, we study more practical *black-box scenarios*, *i.e.*, the attacker crafts $v$ on a *surrogate* model and applies it to an unknown victim model.

**Adversary's Goals.** We consider three DeepSloth variants, (i) the *standard*, (ii) the *universal* and (iii) the *class-universal*. The adversary, in (i) crafts a different $v$ for each $x \in \mathcal{S}$; in (ii) crafts a single $v$ for all $x \in \mathcal{S}$; in (iii) crafts a single $v$ for a target class $i \in M$. Further, although the adversary does not explicitly target it; we observe that DeepSloth usually hurts the accuracy. By modifying the objective function we describe in Sec 4.3, we also experiment with DeepSloth variants that can explicitly preserve or hurt the accuracy, in addition to causing slowdowns.

## 4.2 STANDARD ADVERSARIAL ATTACKS DO NOT CAUSE DELAYS

To motivate DeepSloth, we first evaluate whether previous adversarial attacks have any effect on the efficacy of multi-exit models. These attacks add imperceptible perturbations to a victim's test-time samples to force misclassifications. We experiment with the standard PGD attack (Madry et al., 2017); PGD-avg and PGD-max variants against multi-exit models (Hu et al., 2020) and the Universal Adversarial Perturbation (UAP) attack that crafts a single perturbation for all test samples (Moosavi-Dezfooli et al., 2017). Table 1 summarizes our findings that these attacks, although they hurt the accuracy, fail to cause any meaningful decrease in efficacy. In many cases, we observe that the attacks actually increase the efficacy. These experiments help us to identify the critical elements of the objective function of an attack that decreases the efficacy.

Table 1: **Impact of existing evasion attacks on efficacy.** Each entry shows a model's efficacy (*left*) and accuracy (*right*) when subjected to the respective attack. The multi-exit models are trained on CIFAR-10 and use RAD<5% as their early-exit strategy.

| NETWORK | NO ATTACK | PGD-20 | PGD-20 (AVG.) | PGD-20 (MAX.) | UAP |
|---|---|---|---|---|---|
| **VGG-16** | 0.77 / 89% | 0.79 / 29% | 0.85 / 10% | 0.81 / 27% | 0.71 / 68% |
| **RESNET-56** | 0.52 / 87% | 0.55 / 12% | 0.82 / 1% | 0.70 / 6% | 0.55 / 44% |
| **MOBILENET** | 0.83 / 87% | 0.85 / 14% | 0.93 / 3% | 0.89 / 12% | 0.77 / 60% |

## 4.3 THE DEEPSLOTH ATTACK

**The Layer-Wise Objective Function.** Figure 3 shows that the attacks that only optimize for the final output, *e.g.*, PGD or UAP, do not perturb the model's earlier layer representations. This does not bypass the early-exits, which makes these attacks ineffective for decreasing the efficacy. Therefore, we modify the objective functions of adversarial example-crafting algorithms to incorporate the outputs of all $F_i | i < K$. For crafting $\ell_\infty$, $\ell_2$ and $\ell_1$-bounded perturbations, we adapt the PGD (Madry et al., 2017), the DDN (Rony et al., 2019) and the SLIDE algorithms (Tramèr & Boneh, 2019), respectively. Next, we describe how we modify the PGD algorithm—we modify the others similarly:

$$v^{t+1} = \Pi_{||v||_\infty < \epsilon} \left( v^t + \alpha \, \text{sgn} \left( \nabla_v \sum_{x \in D'} \sum_{0 < i < K} \mathcal{L} \left( F_i \left( x + v \right), \bar{y} \right) \right) \right)$$

Here, $t$ is the current attack iteration; $\alpha$ is the step size; $\Pi$ is the projection operator that enforces $||v||_\infty < \epsilon$ and $\mathcal{L}$ is the cross-entropy loss function. The selection of $\mathcal{D}'$ determines the type of the attack. For the standard variant: $\mathcal{D}' = \{x\}$, *i.e.*, a single test-time sample. For the universal variant: $\mathcal{D}' = \mathcal{D}$, *i.e.*, the whole training set. For the class-universal variant against the target class $i \in M$: $\mathcal{D}' = \{(x, y) \in \mathcal{D} | y = i\}$, *i.e.*, the training set samples from the $i^{th}$ class. Finally, $\bar{y}$ is the target label distribution our objective pushes $F_i(x)$ towards. Next, we explain how we select $\bar{y}$.

**Pushing $F_i(x)$ Towards a Uniform Distribution.** Despite including all $F_i$, attacks such as PGD-avg and PGD-max (Hu et al., 2020) still fail to decrease efficacy. How these attacks select $\bar{y}$ reflects their goal of causing misclassifications and, therefore, they trigger errors in early-exits, *i.e.*, $\text{argmax}_{j \in M} F_i^{(j)}(x) = \bar{y} \neq y$. However, as the early-exits still have high confidence, or low entropy, the model still stops its computation early. We select $\bar{y}$ as a *uniform distribution* over the class labels, *i.e.*, $\bar{y}^{(i)} = 1/m$. This ensures that $(x + v)$ bypasses common stopping criteria as a uniform $F_i(x)$ has both the lowest confidence and the highest entropy.

## 5 EMPIRICAL EVALUATION

Here, we present the results for $\ell_\infty$ DeepSloth against two SDNs—VGG-16 and MobileNet-based—and against the MSDNets. In the Appendix, we report the hyperparameters; the $\ell_1$ and $\ell_2$ attacks; the results on ResNet-56-based SDNs; the cost of the attacks; and some perturbed samples. Overall, we observe that $\ell_\infty$-bounded perturbations are more effective for slowdowns. The optimization challenges might explain this, as $\ell_1$ and $\ell_2$ attacks are usually harder to optimize (Carlini & Wagner, 2017; Tramèr & Boneh, 2019). Unlike objectives for misclassifications, the objective for slowdowns involves multiple loss terms and optimizes over all the output logits.

### 5.1 WHITE-BOX SCENARIOS

**Perturbations Eliminate Early-Exits.** Table 2 (*second* segment) shows that the victim models have $\sim 0$ efficacy on the samples perturbed by DeepSloth. Across the board, the attack makes the early-exit completely ineffective and force the victim models to forward all input samples till the end. Further, DeepSloth also drops the victim's accuracy by 75–99%, comparable to the PGD attack. These results give an answer to our main research question: *the multi-exit mechanisms are vulnerable and their benefits can be maliciously offset by adversarial input perturbations*. In particular, as SDN modification mitigates overthinking in standard, non-adaptive DNNs (Kaya et al., 2019), DeepSloth also leads SDN-based models to overthink on almost all samples by forcing extra computations.

Note that crafting a single perturbation requires multiple back-propagations through the model and more floating points operations (*FLOPs*) than the forward pass. The high cost of crafting, relative to the computational damage to the victim, might make this vulnerability unattractive for the adversary. In the next sections, we highlight scenarios where this vulnerability might lead to practical exploitation. First, we show that in an IoT-like scenarios, the input transmission is a major bottleneck and DeepSloth can exploit it. Second, we evaluate universal DeepSloth attacks that enable the adversary to craft the perturbation only once and reuse it on multiple inputs.

**Attacking an IoT Scenario.** Many IoT scenarios, *e.g.*, health monitoring for elderly (Park et al., 2017), require collecting data from edge devices and making low-latency inferences on this data. However, complex deep learning models are impractical for low-power edge devices, such as an Arduino, that are common in the IoT scenarios (Chen & Ran, 2019). For example, on standard hardware, an average inference takes MSDNet model on Tiny ImageNet 35M FLOPs and $\sim$10ms.

A potential solution is sending the inputs from the edge to a cloud model, which then returns the prediction. Even in our optimistic estimate with a nearby AWS EC2 instance, this back-and-forth introduces $\sim$11ms latency per inference. Model partitioning alleviates this bottleneck by splitting a multi-exit model into two; deploying the small first part at the edge and the large second part at the cloud (De Coninck et al., 2015). The edge part sends an input only when its prediction does not meet

Table 2: **The effectiveness of $\ell_\infty$ DeepSloth.** 'RAD<5,15%' columns list the results in each early-exit setting. Each entry includes the model's efficacy (*left*) and accuracy (*right*). The class-universal attack's results are an average of 10 classes. 'TI': Tiny ImageNet and 'C10': CIFAR-10.

| NETWORK | MSDNET | | VGG16 | | MOBILENET | |
|---|---|---|---|---|---|---|
| SET. | RAD<5% | RAD<15% | RAD<5% | RAD<15% | RAD<5% | RAD<15% |
| **BASELINE (NO ATTACK)** | | | | | | |
| C10 | 0.89 / 85% | 0.89 / 85% | 0.77 / 88% | 0.89 / 79% | 0.83 / 87% | 0.92 / 79% |
| TI | 0.64 / 55% | 0.83 / 50% | 0.39 / 57% | 0.51 / 52% | 0.42 / 57% | 0.59 / 51% |
| **DEEPSLOTH** | | | | | | |
| C10 | 0.06 / 17% | 0.06 / 17% | 0.01 / 13% | 0.04 / 16% | 0.01 / 12% | 0.06 / 16% |
| TI | 0.06 / 7% | 0.06 / 7% | 0.00 / 2% | 0.01 / 2% | 0.02 / 6% | 0.04 / 6% |
| **UNIVERSAL DEEPSLOTH** | | | | | | |
| C10 | 0.85 / 65% | 0.85 / 65% | 0.62 / 65% | 0.86 / 60% | 0.73 / 61% | 0.90 / 59% |
| TI | 0.58 / 46% | 0.81 / 41% | 0.31 / 47% | 0.44 / 44% | 0.33 / 47% | 0.51 / 43% |
| **CLASS-UNIVERSAL DEEPSLOTH** | | | | | | |
| C10 | 0.82 / 32% | 0.82 / 32% | 0.47 / 35% | 0.78 / 33% | 0.60 / 30% | 0.85 / 27% |
| TI | 0.41 / 21% | 0.71 / 17% | 0.20 / 28% | 0.33 / 27% | 0.21 / 27% | 0.38 / 25% |

the stopping criteria. For example, the first early-exit of MSDNets sends only 5% and 67% of all test samples, on CIFAR-10 and Tiny ImageNet, respectively. This leads to a lower average latency per inference, *i.e.*, from 11ms down to 0.5ms and 7.4ms, respectively.

The adversary we study uses DeepSloth perturbations to force the edge part to send all the input samples to the cloud. For the victim, we deploy MSDNet models that we split into two parts at their first exit point. Targeting the first part with DeepSloth forces it to send 96% and 99.97% of all test samples to the second part. This increases average inference latency to ~11ms and invalidates the benefits of model partitioning. In this scenario, perturbing each sample takes ~2ms on a Tesla V-100 GPU, *i.e.*, the time adversary spends is amplified by 1.5-5× as the victim's latency increase.

**Reusable Universal Perturbations.** The universal attacks, although limited, are a practical as the adversary can reuse the same perturbation indefinitely to cause minor slowdowns. Table 2 (*third* segment) shows that they decrease the efficacy by 3–21% and the accuracy by 15–25%, over the baselines. Having a less conservative early-exit strategy, *e.g.*, RAD<15%, increases the resilience to the attack at the cost of accuracy. Further, MSDNets are fairly resilient with only 3–9% efficacy drop; whereas SDNs are more vulnerable with 12–21% drop. The attack is also slightly more effective on the more complex task, Tiny ImageNet, as the early-exits become easier to bypass. Using random noise as a baseline, i.e., $v \sim U^d(-\epsilon, \epsilon)$, we find that at most it decreases the efficacy by ~3%.

In the universal attack, we observe a phenomenon: it pushes the samples towards a small subset of all classes. For example, ~17% of the perturbed samples are classified into the 'bird' class of CIFAR-10; up from ~10% for the clean samples. Considering certain classes are distant in the feature space, *e.g.*, 'truck' and 'bird'; we expect the class-universal variant to be more effective. The results in Table 2 (*fourth* segment) confirm our intuition. We see that this attack decreases the baseline efficacy by 8–50% and the accuracy by 50–65%. We report the average results across multiple classes; however, we observe that certain classes are slightly more vulnerable to this attack.

**Feature Visualization of DeepSloth.** In Figure 3, to shed light on how DeepSloth differs from prior attacks, *e.g.*, PGD and PGD-avg, we visualize a model's hidden block (layer) features on the original and perturbed test-time samples. We observe that in an earlier block (*left* panel), DeepSloth seems to disrupt the original features slightly more than the PGD attacks. Leaving earlier representations intact prevents PGDs from bypassing the early-exits. The behaviors of the attacks diverge in the middle blocks (*middle* panel). Here, DeepSloth features remain closer to the original features than prior attacks. The significant disruption of prior attacks leads to high-confidence misclassifications and fails to bypass early-exits. In the later block (*right* panel), we see that the divergent behavior persists.

**Preserving or Hurting the Accuracy with DeepSloth.** Here, we aim to answer whether DeepSloth can be applied when the adversary explicitly aims to cause or prevent misclassifications, while still

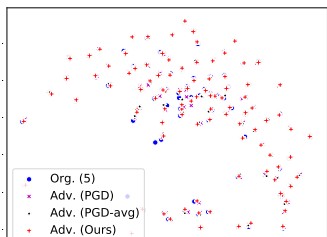 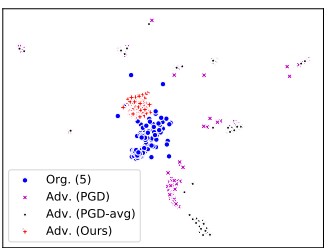 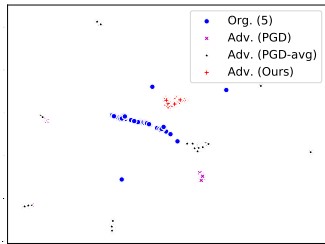

Figure 3: **Visualising features against attacks using UMAP.** VGG-16's 3rd (*left*), 8th (*middle*), and 14th (*right*) hidden block features on CIFAR-10's 'dog' class (Best viewed in color, zoomed in).

causing slowdowns. Our main threat model has no explicit goal regarding misclassifications that hurt the user of the model, *i.e.*, who consumes the output of the model. Whereas, slowdowns additionally hurt the executor or the owner of the model through the computations and latency increased at the cloud providers. In some ML-in-the-cloud scenarios, where these two are different actors, the adversary might aim to target only the executor or both the executor and the user. To this end, we modify our objective function to push $F_i(x)$ towards a slightly non-uniform distribution, favoring either the ground truth label for preventing misclassifications or a wrong label for causing them. We test this idea on our VGG-16-based SDN model on CIFAR-10 in RAD<5% setting. We see that DeepSloth for preserving the accuracy leads to 81% accuracy with 0.02 efficacy and DeepSloth for hurting the accuracy leads to 4% accuracy with 0.01 efficacy—the original DeepSloth led to 13% accuracy with 0.01 efficacy. These results show the flexibility of DeepSloth and how it could be modified depending on the attacker's goals.

## 5.2 TOWARDS A BLACK-BOX ATTACK: TRANSFERABILITY OF DEEPSLOTH

Transferability of adversarial examples imply that they can still hurt a model that they were not crafted on (Tramèr et al., 2017a; Liu et al., 2017). Even though white-box attacks are important to expose the vulnerability, black-box attacks, by requiring fewer assumptions, are more practical. Here, on four distinct scenarios, we investigate whether DeepSloth is transferable. Based on the scenario's constraints, we (i) train a *surrogate* model; (ii) craft the DeepSloth samples on it; and (iii) use these samples on the victim. We run these experiments on CIFAR-10 in the RAD<5%.

**Cross-Architecture.** First, we relax the assumption that the attacker knows the victim architecture. We evaluate the transferability between a VGG-16-based SDN an an MSDNet—all trained using the same $\mathcal{D}$. We find that the samples crafted on the MSDNet can slowdown the SDN: reducing its efficacy to 0.63 (from 0.77) and accuracy to 78% (from 88%). Interestingly, the opposite seems not to be the case: on the samples crafted against the SDN, the MSDNet still has 0.87 efficacy (from 0.89) and 73% accuracy (from 85%). This hints that DeepSloth transfers if the adversary uses an effective multi-exit models as the surrogate.

**Limited Training Set Knowledge.** Second, we relax the assumption that the attacker knows the victim's training set, $\mathcal{D}$. Here, the attacker only knows a random portion of $\mathcal{D}$, i.e., 10%, 25%, and 50%. We use VGG-16 architecture for both the surrogate and victim models. In the 10%, 25% and 50% settings, respectively, the attacks reduce the victim's efficacy to 0.66, 0.5, 0.45 and 0.43 (from 0.77); its accuracy to 81%, 73%, 72% and 74% (from 88%). Overall, the more limited the adversary's $\mathcal{D}$ is, the less generalization ability the surrogate has and the less transferable the attacks are.

**Cross-Domain.** Third, we relax the assumption that the attacker exactly knows the victim's task. Here, the attacker uses $\mathcal{D}_f$ to train the surrogate, different from the victim's $\mathcal{D}$ altogether. We use a VGG-16 on CIFAR-100 as the surrogate and attack a VGG-16-based victim model on CIFAR-10. This transfer attack reduces the victim's efficacy to 0.63 (from 0.77) and its accuracy to 83% (from 88%). We see that the cross-domain attack might be more effective than the limited $\mathcal{D}$ scenarios. This makes DeepSloth particularly dangerous as the attacker, without knowing the victim's $\mathcal{D}$, can collect a similar dataset and still slowdown the victim. We hypothesize the transferability of earlier layer features in CNNs (Yosinski et al., 2014) enables the perturbations attack to transfer from one domain to another, as long as they are similar enough.

**Cross-Mechnanism.** Finally, we test the scenario where the victim uses a completely different mechanism than a multi-exit architecture to implement input adaptiveness, *i.e.*, SkipNet (Wang et al., 2018). A SkipNet, a modified residual network, selectively skips convolutional blocks based on the activations of the previous layer and, therefore, does not include any internal classifiers. We use a pre-trained SkipNet on CIFAR-10 that reduces the average computation for each input sample by $\sim$50% over an equivalent ResNet and achieves $\sim$94% accuracy. We then feed DeepSloth samples crafted on a MSDNet to this SkipNet, which reduces its average computational saving to $\sim$32% (36% less effective) and its accuracy to 37%. This result suggests that the two different mechanisms have more in common than previously known and might share the vulnerability. We believe that understanding the underlying mechanisms through which adaptive models save computation is an important research question for future work.

## 6    STANDARD ADVERSARIAL TRAINING IS NOT A COUNTERMEASURE

In this section, we examine whether a defender can adapt a standard countermeasure against adversarial perturbations, adversarial training (AT) (Madry et al., 2018), to mitigate our attack. AT decreases a model's sensitivity to perturbations that significantly change the model's outputs. While this scheme is effective against adversarial examples that aim to trigger misclassifications; it is unclear whether using our DeepSloth samples for AT can also robustify a multi-exit model against slowdown attacks.

To evaluate, we train our multi-exit models as follows. We first take a *base* network—VGG-16—and train it on CIFAR-10 on PGD-10 adversarial examples. We then convert the resulting model into a multi-exit architecture, using the modification from (Kaya et al., 2019). During this conversion, we adversarially train individual exit points using PGD-10, PGD-10 (avg.), PGD-10 (max.), and DeepSloth; similar to (Hu et al., 2020). Finally, we measure the efficacy and accuracy of the trained models against PGD-20, PGD-20 (avg.), PGD-20 (max.), and DeepSloth, on CIFAR-10's test-set.

Table 3: **Evaluating adversarial training against slowdown attacks.** Each entry includes the model's efficacy score (*left*) and accuracy (*right*). Results are on CIFAR-10, in the RAD<5% setting.

| ADV. TRAINING | NO ATTACK | PGD-20 | PGD-20 (AVG.) | PGD-20 (MAX.) | DEEPSLOTH |
|---|---|---|---|---|---|
| **UNDEFENDED** | 0.77 / 89% | 0.79 / 29% | 0.85 / 10% | 0.81 / 27% | **0.01** / 13% |
| **PGD-10** | 0.61 / 72% | 0.55 / 38% | 0.64 / 23% | 0.58 / 29% | **0.33** / 70% |
| **PGD-10** (AVG.) | 0.53 / 72% | 0.47 / 36% | 0.47 / 35% | 0.47 / 35% | **0.32** / 70% |
| **PGD-10** (MAX.) | 0.57 / 72% | 0.51 / 37% | 0.54 / 30% | 0.52 / 34% | **0.32** / 70% |
| **OURS** | 0.74 / 72% | 0.71 / 38% | 0.82 / 14% | 0.77 / 21% | **0.44** / 67% |
| **OURS + PGD-10** | 0.61 / 73% | 0.55 / 38% | 0.63 / 23% | 0.58 / 28% | **0.33** / 70% |

Our results in Table 3 verify that AT provides resilience against all PGD attacks. Besides, AT provides some resilience to our attack: DeepSloth reduces the efficacy to $\sim$0.32 on robust models vs. 0.01 on the undefended one. However, we identify a trade-off between the robustness and efficiency of multi-exits. Compared to the undefended model, on clean samples, we see that robust models have lower efficacy—0.77 vs. 0.53$\sim$0.61. We observe that the model trained only with our DeepSloth samples (Ours) can recover the efficacy on both the clean and our DeepSloth samples, but this model loses its robustness against PGD attacks. Moreover, when we train a model on both our DeepSloth samples and PGD-10 (Ours + PGD-10), the trained model suffers from low efficacy. Our results imply that a defender may require an out-of-the-box defense, such as flagging the users whose queries bypass the early-exits more often than clean samples for which the multi-exit network was calibrated.

## 7    CONCLUSIONS

This work exposes the vulnerability of input-adaptive inference mechanisms against adversarial slowdowns. As a vehicle for exploring this vulnerability systematically, we propose DeepSloth, an attack that introduces imperceptible adversarial perturbations to test-time inputs for offsetting the computational benefits of multi-exit inference mechanisms. We show that a white-box attack, which perturbs each sample individually, eliminates any computational savings these mechanisms provide. We also show that it is possible to craft universal slowdown perturbations, which can be

reused, and transferable samples, in a black-box setting. Moreover, adversarial training, a standard countermeasure for adversarial perturbations, is not effective against DeepSloth. Our analysis suggests that slowdown attacks are a realistic, yet under-appreciated, threat against adaptive models.

## ACKNOWLEDGMENT

We thank the anonymous reviewers for their feedback. This research was partially supported by the Department of Defense.

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

## A   MOTIVATING EXAMPLES

Here, we discuss two exemplary scenarios where an adversary can exploit the slowdown attacks.

- **(Case 1) Attacks on cloud-based IoT applications.** In most cases, cloud-based IoT applications, such as Apple Siri, Google Now, or Microsoft Cortana, run their DNN inferences in the cloud. This cloud-only approach puts all the computational burden on cloud servers and increases the communications between the servers and IoT devices. In consequence, recent work (Kang et al., 2017; Li et al., 2018; Zhou et al., 2019) utilizes multi-exit architectures for bringing computationally expensive models, *e.g.* language models (Zhou et al., 2020; Hou et al., 2020), in the cloud to IoT (or mobile) devices. They split a multi-exit model into two partitions and deploy each of them to a server and IoT devices, respectively. Under this scheme, the cloud server only takes care of complex inputs that the shallow partition cannot correctly classify at the edge. As a result, one can reduce the computations in the cloud and decrease communications between the cloud and edge.

  On the other hand, our adversary, by applying human-imperceptible perturbations, can convert simple inputs into complex inputs. These adversarial inputs will bypass early-exits and, as a result, reduce (or even offset) the computational and communication savings provided by prior work.

  Here, a defender may deploy DoS defenses such as firewalls or rate-limiting. In this setting, the attacker may not cause DoS because defenses keep the communications between the server and IoT devices under a certain-level. Nevertheless, the attacker still increases: (i) the computations at the edge (by making inputs skip early-exits) and (ii) the number of samples that cloud servers process. Recall that a VGG-16 SDN model classifies  90% of clean CIFAR-10 instances correctly at the first exit. If the adversarial examples crafted by the attacker bypass only the first exit, one can easily increase the computations on IoT devices and make them send requests to the cloud.

- **(Case 2) Attacks on real-time DNN inference for resource- and time-constrained scenarios.** Recent work on the real-time systems (Hu et al., 2019; Jiang et al., 2019) harnesses multi-exit architectures and model partitioning as a solution to optimize real-time DNN inference for resource- and time-constrained scenarios. Hu et al. (2019) showed a real-world prototype of an optimal model partitioning, which is based on a self-driving car video dataset, can improve latency and throughput of partitioned models on the cloud and edge by $6.5$–$14\times$, respectively.

  However, the prior work does not consider the danger of slowdown attacks; our threat model has not been discussed before in the literature. Our results in Sec 5 suggest that slowdown can be induced adversarially, potentially violating real-time guarantees. For example, our attacker can force partitioned models on the cloud and edge to use maximal computations for inference. Further, the same adversarial examples also require the inference results from the model running on the cloud, which potentially increases the response time of the edge devices by $1.5$–$5\times$. Our work showed that multi-exit architectures should be used with caution in real-time systems.

## B   HYPERPARAMETERS

In our experiments, we use the following hyperparameters to craft adversarial perturbations.

$\ell_\infty$**-based DeepSloth.** We find that $\ell_\infty$-based DeepSloth does *not* require careful tuning. For the standard attack, we set the total number of iterations to 30 and the step size to $\alpha = 0.002$. For the modified attacks for hurting or preserving the accuracy, we set the total number of iterations to 75 and the step size to $\alpha = 0.001$. We compute the standard perturbations using the entire 10k test-set samples in CIFAR-10 and Tiny Imagenet. For the universal variants, we set the total number of iterations to 12 and reduce the initial step size of $\alpha = 0.005$ by a factor of 10 every 4 iterations. To compute a universal perturbation, we use randomly chosen 250 (CIFAR-10) and 200 (Tiny Imagenet) training samples.

$\ell_2$**-based DeepSloth.** For both the standard and universal attacks, we set the total number of iterations to 550 and the step size $\gamma$ to 0.1. Our initial perturbation has the $\ell_2$-norm of 1.0. Here, we use the same number of samples for crafting the standard and universal perturbations as the $\ell_\infty$-based attacks.

$\ell_1$**-based DeepSloth.** For our standard $\ell_1$-based DeepSloth, we set the total number of iterations to 250, the step size $\alpha$ to 0.5, and the gradient sparsity to 99. For the universal variants, we reduce the total number of iterations to 100 and set the gradient sparsity to 90. Other hyperparameters remain the same. We use the same number of samples as the $\ell_\infty$-based attacks, to craft the perturbations.

## C  EMPIRICAL EVALUATION OF $\ell_1$ AND $\ell_2$ DEEPSLOTH

Table 4 and Table 5 shows the effectiveness of $\ell_1$-based and $\ell_2$-based DeepSloth attacks, respectively.

Table 4: **The effectiveness of $\ell_1$ DeepSloth.** 'RAD<5,15%' columns list the results in each early-exit setting. Each entry includes the model's efficacy score (*left*) and accuracy (*right*). The class-universal attack's results are an average of 10 classes. 'TI' is Tiny Imagenet and 'C10' is CIFAR-10.

| NETWORK | MSDNET | | VGG16 | | MOBILENET | |
|---|---|---|---|---|---|---|
| SET. | RAD<5% | RAD<15% | RAD<5% | RAD<15% | RAD<5% | RAD<15% |
| | BASELINE (NO ATTACK) | | | | | |
| C10 | 0.89 / 85% | 0.89 / 85% | 0.77 / 89% | 0.89 / 79% | 0.83 / 87% | 0.92 / 79% |
| TI | 0.64 / 55% | 0.83 / 50% | 0.39 / 57% | 0.51 / 52% | 0.42 / 57% | 0.59 / 51% |
| | DEEPSLOTH | | | | | |
| C10 | 0.36 / 51% | 0.35 / 51% | 0.12 / 36% | 0.34 / 45% | 0.18 / 41% | 0.49 / 53% |
| TI | 0.23 / 37% | 0.51 / 40% | 0.08 / 22% | 0.15 / 25% | 0.08 / 33% | 0.19 / 35% |
| | UNIVERSAL DEEPSLOTH | | | | | |
| C10 | 0.89 / 83% | 0.89 / 83% | 0.75 / 85% | 0.88 / 75% | 0.82 / 85% | 0.92 / 77% |
| TI | 0.64 / 55% | 0.83 / 50% | 0.38 / 57% | 0.51 / 52% | 0.41 / 57% | 0.59 / 51% |
| | CLASS-UNIVERSAL DEEPSLOTH | | | | | |
| C10 | 0.88 / 73% | 0.88 / 73% | 0.69 / 78% | 0.86 / 67% | 0.76 / 74% | 0.89 / 65% |
| TI | 0.64 / 54% | 0.83 / 49% | 0.39 / 59% | 0.50 / 58% | 0.41 / 60% | 0.58 / 53% |

Table 5: **The effectiveness of $\ell_2$ DeepSloth.** 'RAD<5,15%' columns list the results in each early-exit setting. Each entry includes the model's efficacy score (*left*) and accuracy (*right*). The class-universal attack's results are an average of 10 classes. 'TI' is Tiny Imagenet and 'C10' is CIFAR-10.

| NETWORK | MSDNET | | VGG16 | | MOBILENET | |
|---|---|---|---|---|---|---|
| SET. | RAD<5% | RAD<15% | RAD<5% | RAD<15% | RAD<5% | RAD<15% |
| | BASELINE (NO ATTACK) | | | | | |
| C10 | 0.89 / 85% | 0.89 / 85% | 0.77 / 89% | 0.89 / 79% | 0.83 / 87% | 0.92 / 79% |
| TI | 0.64 / 55% | 0.83 / 50% | 0.39 / 57% | 0.51 / 52% | 0.42 / 57% | 0.59 / 51% |
| | DEEPSLOTH | | | | | |
| C10 | 0.52 / 64% | 0.52 / 64% | 0.22 / 60% | 0.45 / 62% | 0.23 / 46% | 0.48 / 55% |
| TI | 0.24 / 42% | 0.52 / 44% | 0.13 / 35% | 0.21 / 36% | 0.12 / 38% | 0.25 / 40% |
| | UNIVERSAL DEEPSLOTH | | | | | |
| C10 | 0.89 / 81% | 0.89 / 81% | 0.75 / 87% | 0.88 / 76% | 0.81 / 84% | 0.92 / 76% |
| TI | 0.63 / 54% | 0.82 / 48% | 0.38 / 56% | 0.51 / 52% | 0.41 / 56% | 0.58 / 51% |
| | CLASS-UNIVERSAL DEEPSLOTH | | | | | |
| C10 | 0.88 / 73% | 0.88 / 73% | 0.71 / 81% | 0.86 / 70% | 0.76 / 76% | 0.89 / 66% |
| TI | 0.64 / 53% | 0.83 / 49% | 0.38 / 57% | 0.50 / 57% | 0.41 / 58% | 0.58 / 53% |

Our results show that the $\ell_1$- and $\ell_2$-based attacks are less effective than the $\ell_\infty$-based attacks. In contrast to the $\ell_\infty$-based attacks that eliminate the efficacy of victim multi-exit models, the $\ell_1$- and $\ell_2$-based attacks reduce the efficacy of the same models by 0.24~0.65. Besides, the accuracy drops caused by $\ell_1$- and $\ell_2$-based attacks are in 6~21%, smaller than that of $\ell_\infty$-based DeepSloth (75~99%). Moreover, we see that the universal variants of $\ell_1$- and $\ell_2$-based attacks can barely reduce the efficacy of multi-exit models—they decrease the efficacy up to 0.08 and the accuracy by 12%.

## D    EMPIRICAL EVALUATION OF DEEPSLOTH ON RESNET56

Table 6 shows the the effectiveness of our DeepSloth attacks on ResNet56-base models.

Table 6: **The effectiveness of DeepSloth on the ResNet-based models.** 'RAD<5,15%' columns list the results in each early-exit setting. Each entry includes the model's efficacy score (*left*) and accuracy (*right*). The class-universal attack's results are an average of 10 classes.

| NETWORK | RESNET ($\ell_\infty$) | | RESNET ($\ell_1$) | | RESNET ($\ell_2$) | |
|---|---|---|---|---|---|---|
| SET. | RAD<5% | RAD<15% | RAD<5% | RAD<15% | RAD<5% | RAD<15% |
| | | BASELINE (NO ATTACK) | | | | |
| C10 | 0.52 / 87% | 0.69 / 80% | 0.52 / 87% | 0.69 / 80% | 0.51 / 87% | 0.69 / 80% |
| TI | 0.25 / 51% | 0.39 / 46% | 0.25 / 51% | 0.39 / 46% | 0.25 / 51% | 0.39 / 46% |
| | | DEEPSLOTH | | | | |
| C10 | 0.00 / 19% | 0.01 / 19% | 0.05 / 43% | 0.18 / 47% | 0.06 / 45% | 0.17 / 48% |
| TI | 0.00 / 7% | 0.01 / 7% | 0.04 / 27% | 0.10 / 28% | 0.05 / 34% | 0.13 / 35% |
| | | UNIVERSAL DEEPSLOTH | | | | |
| C10 | 0.35 / 63% | 0.59 / 60% | 0.49 / 84% | 0.68 / 75% | 0.48 / 85% | 0.67 / 76% |
| TI | 0.25 / 25% | 0.34 / 37% | 0.25 / 51% | 0.39 / 46% | 0.25 / 51% | 0.38 / 46% |
| | | CLASS-UNIVERSAL DEEPSLOTH | | | | |
| C10 | 0.23 / 33% | 0.48 / 29% | 0.39 / 70% | 0.60 / 61% | 0.39 / 71% | 0.60 / 61% |
| TI | 0.11 / 21% | 0.23 / 18% | 0.23 / 51% | 0.36 / 46% | 0.23 / 50% | 0.36 / 46% |

Our results show that ResNet56-based models are vulnerable to all the $\ell_\infty$, $\ell_2$, and $\ell_1$-based DeepSloth attacks. Using our $\ell_\infty$-based DeepSloth, the attacker can reduce the efficacy of the victim models to 0.00~0.01 and the accuracy by 39~68%. Besides, the $\ell_2$, and $\ell_1$-based attacks also decrease the efficacy to 0.04~0.18 and the accuracy by 11~44%. Compared to the results on MSDNet, VGG16, and MobileNet in Table 4 and 5, the same attacks are more effective. The universal variants decrease the efficacy up to 0.21 and the accuracy up to 24%. In particular, the $\ell_2$, and $\ell_1$-based attacks (on CIFAR-10 models) are effective than the same attacks on MSDNet, VGG16, and MobileNet models.

## E    COST OF CRAFTING DEEPSLOTH SAMPLES

In Table 7, we compare the cost of DeepSloth with other attack algorithms on a VGG16-based CIFAR-10 model—executed on a single Nvidia Tesla-V100 GPU. For the universal DeepSloth, we measure the execution time for crafting a perturbation using one batch (250 samples) of the training set. For the other attacks, we measure the time for perturbing the whole test set of CIFAR-10. Our DeepSloth takes roughly the same time as the PGD and PGD-avg attacks and significantly less time than the PGD-max attack. Our

| ATTACKS | TIME (SEC.) |
|---|---|
| PGD-20 | 38 |
| PGD-20 (AVG.) | 48 |
| PGD-20 (MAX.) | 475 |
| DEEPSLOTH | 44 |
| UNIVERSAL DEEPSLOTH | 2 |

Table 7: Time it takes to craft attacks.

universal DeepSloth takes only 2 seconds (10x faster than DeepSloth) as it only uses 250 samples.

## F    ADVERSARIAL EXAMPLES FROM STANDARD ATTACKS AND DEEPSLOTH

In Figure 4, we visualize the adversarial examples from the PGD, UAP and our DeepSloth attacks.

| | Standard Attacks | | | | DeepSloth | | |
|---|---|---|---|---|---|---|---|
| Original | PGD | PGD (avg.) | PGD (max.) | UAP | Per-sample | Universal | Class-specific |
| | | | | | | | |
| ℓinf of the Perturbations (on Average) | | | | | | | |
| $\ell_{\text{inf}}$ | 0.03 | 0.03 | 0.03 | 0.03 | 0.03 | 0.03 | 0.03 |

Figure 4: **Adversarial examples from the standard and our DeepSloth attacks.** The leftmost column shows the clean images. In the next four columns, we show adversarial examples from PGD, PGD (avg.), PGD (max.), and UAP attacks, respectively. The last four columns include adversarial examples from the three variants of DeepSloth. Each row corresponds to each sample, and the last row contains the average $\ell_{\text{inf}}$-norm of the perturbations over the eight samples in each attack.