# OpenReview forum: "A Panda? No, It's a Sloth: Slowdown Attacks on Adaptive Multi-Exit Neural Network Inference"
_ICLR.cc/2021/Conference — ICLR 2021 Spotlight_

### Official Review · AnonReviewer3 · 2020-10-28
**Good paper with an interesting new adversarial threat model**

**Rating:** 8
**Confidence:** 5

**Review:**

This paper defines a new problem named slowdown adversarial attack to decrease inference efficiency of early multi-exit networks.

Pros:
1. Slowdown attack on efficient-inference networks in an innovative new problem with wide real-world applications.
2. The proposed method, basically pushing the model prediction to uniform distribution, is straightforward and intuitively correct.
3. The universal/class-universal variances of slowdown attack provide more flexibility of the proposed method, and the blackbox experiments show good generalization ability of slowdown attack across models and datasets.
4. Last but not least, the authors show that traditional adversarial training is not a cure for the newly proposed slowdown attack, showing the challenging character of this new threat model.

Comments:
1. Limitation in application scenario: From the prospective of an attacker, one may want to jointly attack inference efficacy and accuracy. Is it possible to design a more generalized setting, where the adversarial attacker can generate images that both slowdown inference and leads to wrong prediction? The current method pushes model predictions to uniform distribution, so it may be ineffective to generate adversarial images to fool classifiers. This is validated by the results, where the slowdown adversarial images have an 5% or 15% lower accuracy compared with clean images.
2. Any discussions on how to conduct slowdown attacks on other efficient-inference models (e.g., SkipNet [1])?
3. Please consider releasing your codes.

[1] SkipNet: Learning Dynamic Routing in Convolutional Networks.

---

> ### Author Response · Authors · 2020-11-21
> **Response to Reviewer 3’s Comments**
>
> We thank the reviewer for constructive feedback. Here, we are happy to provide answers to your questions and concerns. We will also update our paper for clarification.
>
> **[Concerns about the Limitation in Application Scenarios]**
>
> Although in our paper the attacks have no goal regarding the accuracy, with a minor modification, on top of causing slowdowns, they can deliberately hurt the accuracy (as per Reviewer 3’s comment) or preserve the accuracy (as per Reviewer 4’s comment). The original attack pushes the model’s outputs to a uniform distribution and these modified attacks push them to a slightly non-uniform one, favoring a wrong label to hurt the accuracy or the ground truth label to preserve the accuracy.
>
> Against the first attack (on CIFAR-10, VGG16), the victim model achieves 4% accuracy and 0.01 efficacy. Against the second attack, the victim model achieves 81% accuracy and 0.02 efficacy (compared with 13% accuracy and 0.01 efficacy, for the original attack).
>
> These initial results show the flexibility of DeepSloth and how it could be modified to preserve or hurt the accuracy, depending on the attacker’s goals. We will update our manuscript to discuss these results as well.
>
> **[Discussion about Attacking SkipNet]**
>
> SkipNet and the architectures we studied---MSDNets and SDNs---implement vastly different adaptive methods. SkipNet uses reinforcement learning to learn when to skip immediate layers, whereas the others learn stopping criteria to stop the computation at an immediate layer altogether.
>
> We could directly apply our attack to SkipNet by attaching early-exit classifiers at its immediate layers. However, instead, by using a transfer attack from MSDNets, we tested whether these two different methods share an unknown mechanism to save computation, and therefore share a vulnerability.
>
> We use a pre-trained SkipNet on CIFAR-10 that saves ~50% computation on an average sample while achieving 94% accuracy. On the DeepSloth samples we crafted against MSDNets, this SkipNet only saves ~32% computation (36% less effective) and achieves 37% accuracy (60% less accurate). Overall, the samples that slowdown MSDNets are also effective against SkipNet.
>
> This surprising result suggests that these methods have more in common than previously known. The underlying mechanisms through which adaptive models save computation, and the transferability of DeepSloth attacks, are interesting questions for future research. We thank the reviewer for suggesting this investigation.
>
> **[Source Code Release]**
>
> In order to promote reproducibility, we will release our source code for the key experiments. We will include an anonymized link to our repository in the updated version of our paper.

---

### Official Review · AnonReviewer1 · 2020-10-28
**Lacking motivation and novelty**

**Rating:** 3
**Confidence:** 3

**Review:**

This paper aims to indicate a new direction to conduct an adversarial attack: to delay the inference time/raise the computational costs of a ''multi-exit'' model.

However, both the motivation and novelty are lacking, I will argue for the rejection if other reviewers wish to accept it.

From the perspective of the industry:
1. We do not actually use the so-called ''multi-exit'' architectures, because they could be difficult to optimize, quantize, assemble, and so on. For mobile devices, we deploy different ''single-exit'' architectures to meet their computing capacity and balance the performance. And it is even trivial to maintenance these architectures because we have developed automation tools.
2. On the cloud servers, the deployment is redundant, such that the service will still work stably, even all the queries require the maximal flops.
3. After all, it is easy to limit one's Queries Per Second (QPS), and nothing need to be worried about.

From the perspective of academia:
1. This paper only employs the PGD attack and the adversarial training on a specific task, the novelty is lacking.
2. In section 5.2, if one knows the architecture and the training set and the task, how could this configuration being called ''black-box''.

UPDATE:

In the rebuttal, the authors emphasize two facts:
1. There exist such IoT scenarios and real-time systems that employ multi-exit architectures (including those that employ cloud computation.).
2. The slowdown attack is effective in these scenarios.

However, to prove this method works in practice, it is not a simple "1 then 2", you need to show us "1 and 2". That is, you do actually deploy any system described in [1,2,3,4,5,6,7,8], and provide a feasible approach to attack with your method, and report the actual damage caused by your method, and convince the readers the damage is significantly severe compared to the efforts spent for causing the damage.
Otherwise, it is only an application of PGD with a different loss function.

---

> ### Author Response · Authors · 2020-11-21
> **Clarification of Our Novelty and Contributions**
>
> We thank the reviewer for the feedback. Below we clarify the motivation and novelty of our contributions.
>
> **[Comments from the Industry Perspective]**
>
> (1) We want to highlight that companies are working on harnessing multi-exit architectures for faster inference. Intel (Barad and Tang [1]) develops multi-exit architecture to reduce the computational costs of DNNs. Microsoft and Huawei [2, 3] conducted research on reducing the computational complexity of Bidirectional Encoder Representations from Transformers (BERT) for bringing language models to IoT (edge) devices.
>
> (2) We also emphasize that there has been work on deploying multi-exit architectures (with model partitioning) to IoT scenarios and real-time systems [4, 5, 6, 7, 8], which reduces the computational costs (maximal FLOPs) and latency caused by using a single-exit architecture.
>
> (3) Slowdown attacks may be effective even with rate-limiting because they increase the computational costs and the latency by making all the samples bypass exit-points and require computations in the cloud. This attack vector is orthogonal to the attacks that rate-limiting protects against.
>
> **[Comments from the Academic Perspective]**
>
> (1) In addition to PGD, we evaluate the DDN and SLIDE algorithms for crafting ell-1 and ell-2 bounded perturbations (see Appendix C). We also propose DeepSloth, a technique for adapting attacks to a range of adversarial goals, including slowdown and other goals suggested by the reviewers (see our response to Reviewer 3). We evaluate the vulnerability of multi-exit architectures to these attacks on the CIFAR-10 benchmark, in line with the prior work on testing the resilience of adversarially-trained models against adversarial examples [9, 10, 11].
>
> (2) We utilize the term “black-box” in line with the prior research [12, 13]. Specifically, in Sec 5.2, when we study the “transferability” of DeepSloth, we assume the attacker does not know the victim model’s architecture. Moreover, in the same section, we examine cases where the conventional transferability studies [13] did not assume: our attacker does not know the training set (the limited training set knowledge and cross-domain scenarios).
>
> **References**
>
> [1] Barad and Tang, Fast Inference with Early Exit, Intel, https://www.intel.com/content/www/us/en/artificial-intelligence/posts/fast-inference-with-early-exit.html
>
> [2] Zhou et al., Bert Loses Patience: Fast and Robust Inference with Early Exit, NeurIPs, 2020.
>
> [3] Hou et al., DynaBERT: Dynamic BERT with Adaptive Width and Depth, NeurIPs, 2020.
>
> [4] Kang et al., Neurosurgeon: Collaborative Intelligence Between the Cloud and Mobile Edge, ASPLOS, 2017.
>
> [5] Li et al., Edge Intelligence: On-Demand Deep Learning Model Co-Inference with Device-Edge Synergy, SIGCOMM, 2018
>
> [6] Zhou et al., Distributing Deep Neural Networks with Containerized Partitions at the Edge, HotEdge, 2019.
>
> [7] Hu et al., Dynamic Adaptive DNN Surgery for Inference Acceleration on the Edge, INFOCOM, 2019.
>
> [8] Jiang et al., Achieving Super-Linear Speedup across Multi-FPGA for Real-Time DNN Inference, ACM Transaction on Embedded Computing Systems, 2019.
>
> [9] Ali et al., Adversarial Training for Free!, NeurIPs 2019.
>
> [10] Wong et al., Fast Is Better Than Free: Revisiting Adversarial Training, ICLR, 2020.
>
> [11] Zhang et al., Theoretically Principled Trade-off between Robustness and Accuracy, ICML, 2019.
>
> [12] Tramer et al., Adversarial Training and Robustness for Multiple Perturbations, NeurIPs, 2019.
>
> [13] Liu et al., Delving into Transferable Adversarial Examples and Black-box Attacks, ICLR 2017.

---

### Official Review · AnonReviewer2 · 2020-10-28
**Detailed investigation of a new threat model**

**Rating:** 6
**Confidence:** 3

**Review:**


Summary of the paper
=================

This paper studies a new category of adversarial attacks, i.e., attackers that try to slow-down multi-exit DNNs using adversarial examples. The paper extended adversarial attacks to perform the slow-down attack and showed that the attacks could slow-down multi-exit DNNs by 1.5x - 5.0x. Additionally, the paper experimentally answers many questions such as (1) the effectiveness of adversarial training against the attack, (2) input-agnostic attack, and (3) cross-architecture/domain transferability.

Strongness
=========

- The paper suggests a new threat model.
- The paper studies the problem from multiple aspects such as new-attacks, the effectiveness of existing and new defenses, and transferability.

Overall, I think the paper suggested a new and interesting problem and also performed sufficient experiments and analyses for the new threat.

Weakness
========

Some of the motivation of the attack is unclear. I am not convinced that attackers are motivated to use the slow-down attack instead of performing DDoS on cloud applications. And I do not think multi-exit models are used on IoT applications if the target is a real-time system (The reviewer is not a specialist in real-time systems, and please correct it if it is wrong). The reviewer believes that more discussion and motivational examples of the new thread model will increase this paper's impact.

Clarity
=====

This paper is well organized and clear.

Relation to prior work
================

To the best of the reviewer's knowledge, the paper is correctly placed in the literature. However, it is not likely but possible that the reviewer is missing some important prior work.

---

> ### Author Response · Authors · 2020-11-21
> **Response to Reviewer 2’s Comments**
>
> We thank the reviewer for constructive feedback. We provide answers to your questions and concerns, and we will update our paper for clarification.
>
> **(1) Concerns about the Motivation of Our Attacks**
>
> **[DoS Attacks on Cloud-based IoT Applications]**
>
> We agree with the reviewer that an adversary may choose other means to inflict DoS attacks on the cloud-based IoT applications [1, 2, 3]. However, slowdown attacks can still increase the computational costs and the latency of those IoT applications that have DoS defenses, such as firewalls or rate-limiting.
>
> Suppose that a victim partitions a model into two components and deploys them on a server and IoT devices, respectively, with defenses. We assume the victim uses a VGG-16 SDN model that classifies ~90% of clean CIFAR-10 samples correctly at the first exit, in the benign settings. Here, the attacker may not cause DDoS since defenses limit the communication between the server and clients under a certain-level. Nevertheless, our attack still increases: (1) the computations at the edge (by making inputs skip early-exits) and (2) the number of samples that cloud servers are processing.
>
> **[Concerns about Attacking Real-time Systems]**
>
> We clarify that recent work on the real-time systems [4, 5] utilizes multi-exit architectures (and model partitioning) as a solution to optimize real-time DNN inference for resource- and time-constrained scenarios. However, the prior work does not consider the danger of slowdown attacks, as our threat model has not been discussed before in the literature. We agree with the reviewer that multi-exit architectures should be used with caution in real-time systems, as our results suggest that slowdown can be induced adversarially, potentially violating real-time guarantees. We thank the reviewer for bringing the discussion about the deployment of multi-exit architectures for real-time systems.
>
> We will update our threat model in Sec 1 and 4.1 and also include our discussion about IoT scenarios and real-time systems in Appendix, as motivating examples.
>
> **(2) A Missing Prior Work**
>
> We identified a missing prior work (Haque et al, 2020 [7]). We will include this paper in our related work, and we will discuss the major differences between our contributions and this work, as outlined below.
>
> **[Our Major Contributions]**
>
> First, we study a different threat model (as Reviewer 3 pointed out). We study an adversary who aims to cause slowdown on multi-exit architectures (i.e., SDNs and MSDNets) by using adversarial input perturbations. There are two consequences of our attack: the attacker (1) can introduce network latency to an infrastructure that utilizes multi-exit architectures and (2) can waste the victim’s computational resources. This is in contrast to the threat model in Haque et al., which presented specific attacks on the energy-efficient DNNs (i.e., AdNN and SkipNet).
>
> Second, We conduct the first systematic study of this new threat model. We emphasize our key contributions that are not in the missing prior work as follows:
>
> (1) Our work identifies the underlying mechanism that makes multi-exit architecture vulnerable to adversarial input perturbations---i.e., confidence-based early classifications. This observation leads to a simple, yet effective attack (see Reviewer 4’s comments) that optimizes the input for the uniform confidence scores at all exits in a multi-exit architecture. Thus, our DeepSloth can be adapted to study the vulnerabilities of other input-adaptive inference mechanisms. In the response to Reviewer 3, we showed our resulting attacks are also effective against SkipNet, although it uses a vastly different adaptive mechanism than the architectures we studied.
>
> (2) We consider a practical attack. The attacks presented in Haque et al. take multiple seconds because of the complex optimization procedure. However, the fact that the attack costs more FLOPs than an increase in FLOPs in the target model makes the attack impractical for a real-world adversary. On the other hand, our DeepSloth can craft an adversarial example in 2.4 milliseconds that can completely eliminate the computational savings of a multi-exit model Moreover, our adversarial example increased network latency up to 5x in an IoT deployment of a multi-exit architecture.
>
> (3) We define a metric---the early-exit efficacy---that can be used to measure the impact of any adversarial input perturbation on multi-exit architectures. Unlike the conventional metrics such as the number of FLOPs that vary by architecture configurations, our metric provides a standardized conservative estimate of the increase/decrease in latency or computations required.
>
> *Continued on the next comment.*

---

> > ### Author Response · Authors · 2020-11-21
> > **Response to Reviewer 2’s Comments - Cont'd**
> >
> > (4) We study the transferability of DeepSloth. Our work considers the attacker who does not know the victim’s architecture, the training data, or the task that the model is trained for. We show that in certain scenarios our attack can still hurt an unknown model, which further makes it more practical for a real-world adversary.
> >
> > (5) Our work also discussed potential defense mechanisms against this vulnerability, by proposing a simple adaptation of adversarial training (AT). We showed that AT provides some resilience to DeepSloth, but the attacker can still reduce the efficacy of a victim model to ~0.32. We also exposed the trade-off between the robustness and efficacy in AT models.
> >
> > **References**
> >
> > [1] Kang et al., Neurosurgeon: Collaborative Intelligence Between the Cloud and Mobile Edge, ASPLOS, 2017.
> >
> > [2] Li et al., Edge Intelligence: On-Demand Deep Learning Model Co-Inference with Device-Edge Synergy, SIGCOMM, 2018
> >
> > [3] Zhou et al., Distributing Deep Neural Networks with Containerized Partitions at the Edge, HotEdge, 2019.
> >
> > [4] Hu et al., Dynamic Adaptive DNN Surgery for Inference Acceleration on the Edge, INFOCOM, 2019
> >
> > [5] Jiang et al., Achieving Super-Linear Speedup across Multi-FPGA for Real-Time DNN Inference, ACM Transaction on Embedded Computing Systems, 2019.
> >
> > [7] Haque et al., ILFO: Adversarial Attack on Adaptive Neural Networks, CVPR, 2020.

---

### Official Review · AnonReviewer4 · 2020-11-02
**Interesting problem, simple yet effective method, good results.**

**Rating:** 7
**Confidence:** 4

**Review:**

This paper studies adversarial attack and defense for adaptive multi-exit network. Adaptive multi-exit network is, by itself, a pretty new and under-studied topic, let alone the adversarial study on top of it. This paper proposes a simple-yet-effective DeepSloth attack based on layerwise loss function. It also proposed an efficacy metric for better evaluation. The experiments are conducted on four multi-exit networks and two datasets: cifar10 and tiny imagenet. In the appendix, there is also evaluation on different norms of attacks. The results on white-box and black-box attack demonstrate the effectiveness: DeepSloth not only hurts the accuracy (also achieved by baselines), but also hurts the efficacy (only achieved by DeepSloth). To reduce the computation burden of performing the attack, the authors did two things: 1) model partitioning in scenario of IOT, and 2) universal attack across a dataset. At the end, the authors adapt AT to multi-exit networks and demonstrate that AT is effective in general and DeepSloth is further helping AT for better robustness.

Overall, I think this paper is pretty complete, containing both attack side and defense side. The proposed method is simple and effective, and the empirical results look good.

My main concern is about the usefulness. The main gain over baseline attack is that DeepSloth also hurts efficacy. In practical use case, two factors lead to bad results to users: 1) inference timeout; 2) wrong prediction. So if we already got wrong prediction, why we care efficacy (which may cause inference timeout)?

Some other minor points.

1. Section 3: metrics. This is novelty of this paper, so I think should be emphasized somewhat.

2. From Appendix, why DeepSloth is half time versus PGD-(avg)?

---

> ### Author Response · Authors · 2020-11-21
> **Response to Reviewer 4’s Comments**
>
> We thank the reviewer for constructive feedback. Here, we provide answers to your questions and concerns, and we will update our paper for clarification.
>
> **(1) Usefulness of Our DeepSloth**
>
> As the reviewer points out, a misclassification already hurts the user (i.e., whoever is consuming the output of the model), even in the absence of a slowdown. However, slowdowns additionally hurt the executor or the owner of the model through the computations and latency increased at the cloud providers. In ML-in-the-cloud scenarios, oftentimes these two are different actors.
>
> Moreover, users can employ standard defenses against adversarial examples, e.g., adversarial training, to mitigate misclassifications. However, we show that these defenses are relatively ineffective against DeepSloth slowdowns. Our work demonstrates a new dimension of multi-exit models’ threat surface even when the model is resistant to adversarial misclassifications.
>
> Further, in our response to Reviewer 3, we show that DeepSloth’s objective function can be modified for scenarios where the attacker aims to cause slowdowns while not triggering a misclassification. In this scenario, the attacker might want to increase the computational burden on the cloud provider or the financial burden on the model’s owner, while not hurting the end-users of the model.
>
> **(2) About Minor Comments**
>
> **[Novelty of Our Metric]**
>
> We thank the reviewer for pointing this out. We designed this metric to standardize the comparisons between different architectures and approaches. We will highlight the novelty of our metric and its use-cases for future work and update our paper.
>
> **[Cost of Crafting PGD-20 (avg.) and DeepSloth]**
>
> We also thank the reviewer for pointing this out. We found that 24ms for crafting our DeepSloth samples is a typo; the time it takes to perturb those samples is 44ms --- the same as the PGD-20 (avg.) attack. We run the sanity checks for the numbers in Table 4 (Appendix) and confirm that the others are correct.

---

### Public Comment · ~Mirazul_Haque1 · 2020-11-10
**Prior work in this direction**

I acknowledge that this is an important work in this direction. However, our work ILFO (https://openaccess.thecvf.com/content_CVPR_2020/html/Haque_ILFO_Adversarial_Attack_on_Adaptive_Neural_Networks_CVPR_2020_paper.html) was published in this year's CVPR and to my knowledge, it is the first work that attacks Adaptive Neural Networks. Scientific improvements can be better understood if comparison with existing research is shown in the paper.

---

> ### Author Response · Authors · 2020-11-21
> **Clarification of Our Key Contributions Different From Your CVPR Paper**
>
> Hello Mirazul Haque,
>
> Thank you for bringing this missing prior work to our attention. We will include your paper in our related work. Here, we would like to highlight the major differences between our contributions and your work. We will also update our manuscript to emphasize these.
>
> **[Our Major Contributions]**
>
> First, **we study a different threat model (as Reviewer 3 pointed out).** We study an adversary who aims to cause slowdown on multi-exit architectures (i.e., SDNs and MSDNets) by using adversarial input perturbations. There are two consequences of our attack: the attacker (1) can introduce network latency to an infrastructure that utilizes multi-exit architectures and (2) can waste the victim’s computational resources. This is in contrast to the threat model in your paper that presented specific attacks on the energy-efficient neural networks (i.e., AdNN and SkipNet).
>
> Second, **we conduct the first systematic study of this new vulnerability (as Reviewer 2 pointed out).** We emphasize our key contributions that are not in the CVPR paper as follows:
>
> (1) Our work identifies the underlying mechanism that makes multi-exit architecture vulnerable to adversarial input perturbations---i.e., confidence-based early classifications. This observation leads to a simple, yet effective attack (see Reviewer 4’s comments) that optimizes the input for the uniform confidence scores at all exits in a multi-exit architecture. Thus, our DeepSloth can be adapted to study the vulnerabilities of other input-adaptive inference mechanisms. In the response to Reviewer 3, we showed our resulting attacks are also effective against SkipNet, although it uses a vastly different adaptive mechanism than the architectures we studied.
>
> (2) We consider a practical attack. The attacks presented in the CVPR paper take multiple seconds because of the complex optimization procedure. However, the fact that the attack costs more FLOPs than an increase in FLOPs in the target model makes the attack impractical for a real-world adversary. On the other hand, our DeepSloth can craft an adversarial example in 2.4 milliseconds that can completely eliminate the computational savings of a multi-exit model Moreover, our adversarial example increased network latency up to 5x in an IoT deployment of a multi-exit architecture.
>
> (3) We define a metric---the early-exit efficacy---that can be used to measure the impact of any adversarial input perturbation on multi-exit architectures. Unlike the conventional metrics such as the number of FLOPs that vary by architecture configurations, our metric provides a standardized conservative estimate of the increase/decrease in latency or computations required.
>
> (4) We study the transferability of DeepSloth. Our work considers the attacker who does not know the victim’s architecture, the training data, or the task that the model is trained for. We show that in certain scenarios our attack can still hurt an unknown model, which further makes it more practical for a real-world adversary.
>
> (5) Our work also discussed potential defense mechanisms against this vulnerability, by proposing a simple adaptation of adversarial training (AT). We showed that AT provides some resilience to DeepSloth, but the attacker can still reduce the efficacy of a victim model to ~0.32. We also exposed the trade-off between the robustness and efficacy in AT models.

---

### Author Response · Authors · 2020-11-25
**Summary of Our Responses and Changes to the Manuscript**

We thank our reviewers again for taking the time to read, evaluate our work, and provide constructive feedback. We have uploaded a revised version of our paper, with edits to address the concerns raised. Here, we summarize our responses and updates below:

**[Reviewer 1]**

We provided our answers to the concerns about the novelty.

**[Reviewer 2]**

We clarified our threat model and discussed motivating examples. We also identified missing prior work (Haque et al., 2020) and discussed the major differences between our contributions and this work.

**[Reviewer 3]**

We presented results on a modified version of our attack for jointly optimizing for misclassification and a slowdown. We also demonstrated that SkipNet, although it is not a multi-exit model, still is vulnerable to the DeepSloth samples crafted on a multi-exit model, MSDNet.

**[Reviewer 4]**

We clarified the benefits of a slowdown attack over standard adversarial examples that cause misclassifications. We also presented results on a modified version of our attack to only cause slowdowns while preserving the accuracy.

**[Errors Fixed]**

(1) We cited a related prior work from CVPR 2020 pointed out by Mirazul Haque.

(2) We fixed the cost of crafting DeepSloth from 24ms to 44ms in Table 4 (per Reviewer 4’s comment).

**[Manuscript updates]**

(1) We included the anonymized link to our source code repository in the Abstract. (per Reviewer 3’ comment).

(2) We included a "Motivating Examples" section in the Appendix where we discuss our threat model and motivating examples (per Reviewer 2's comment).

(3) We included a “Cross-Mechanism Transferability” section under Section 5.2 for our results on SkipNet (per Reviewer 3’s comment).

(4) We included a “Preserving or Hurting the Accuracy with DeepSloth” under Section 5.1 for our results on modified objective functions (per Reviewer 3 and Reviewer 4’s comments).

Please see our replies to each reviewer for our detailed responses to individual points.

---

### Decision · Program_Chairs · 2021-01-07
**Final Decision**

**Decision:**

Accept (Spotlight)

**Comment:**

The paper discusses a new threat model for multi-exit DNNs: attacks against efficiency of inference. The proposed attack increases the inference time of such networks by the factor of 1.5-5, while at the same reducing the accuracy of attacked networks. Unlike classical adversarial examples, the new type of attack cannot be thwarted by adversarial training.

Overall, the paper exhibits a novel contribution, is well written and methodically sound. Its practical motivation is somewhat weak, as it is currently unclear for which applications such attacks may be feasible. However, the novelty of the threat model addressed by this paper makes it an interesting methodical contribution.